# Black Hole Hyperaccretion in Collapsars: A Review

Yun-Feng Wei  and Tong Liu *

Department of Astronomy, Xiamen University, Xiamen 361005, China
* Correspondence: tongliu@xmu.edu.cn

**Abstract:** The collapsar model is widely accepted as one of the standard scenarios for gamma-ray bursts (GRBs). In the massive collapsar scenario, the core will collapse to a black hole (BH) surrounded by a temporary hyperaccretion disk with a very high accretion rate. The newborn BH hyperaccretion system would launch the relativistic jets via neutrino annihilation and Blandford-Znajek (BZ) mechanism. At the initial accretion stage, the accretion disk should be a neutrino-dominated accretion flow (NDAF). If the jets can break out from the envelope and circumstellar medium, then a GRB will be triggered. In this review, we summarize the theoretical progress on the multimessenger astronomy of the BH hyperaccretion in the center of collapsars. The main topics include: jet propagation in collapsar, MeV neutrinos from NDAFs and proto-neutron stars, gravitational waves from collapsars.

**Keywords:** accretion; accretion disks; black hole physics; gamma-ray bursts; general; gravitational waves; neutrinos

## 1. Introduction

Gamma-ray bursts (GRBs) are one of the most luminous explosions in the universe. The production of GRBs requires a small amount of material accelerated to ultrarelativistic speeds and collimated as a jet [1,2]. The duration of GRB is usually defined by the so-called $T_{90}$ (the time interval between the epochs when 5% and 95% of the total fluence is collected by the detector). Based on the observed bimodal distribution of duration, GRBs can be classified into two categories: long-duration GRBs (LGRBs, $T_{90} > 2$ s) and short-duration GRBs (SGRBs, $T_{90} < 2$ s) [3]. Multimessenger observations have suggested that SGRBs originate from binary neutron star (NS) mergers [4], and plausible NS-black hole (BH) mergers, whereas LGRBs originate from core collapse of massive stars [5]. For LGRBs, the most widely accepted model is the collapsar model [6–9].

A massive collapsar is a massive star (>30 $M_\odot$) whose iron core collapses to a BH and has sufficient angular momentum to form a hyperaccretion disk. Three types of collapsars have been investigated in previous works, which are Types I, II, and III. For Type I collapsars, the star collapses and initially forms a proto-NS (PNS), however, it is unable to launch a supernova (SN) shock and then (after ∼1 s) collapses to form a BH [6,8]. In Type II, the BH is formed by fallback after an initial SN shock has been launched [9,10]. The weak outgoing shock cannot eject much of the star, and the subsequent fallback of materials inside the star cause the NS to collapse to a BH. The progenitor star of Type III collapsar is expected to be a low metallicity massive star. For Type III collapsars, massive stars do not form PNSs, but instead quickly collapse into BHs [11,12]. For Type III collapsar from massive Population III stars, the formed BH is very massive (∼300 $M_\odot$). The types of collapsars are mainly determined by the properties of progenitors, including mass, metallicity, rotation, mass loss, etc. Although their evolutionary paths are different, all three types of collapsars generate BH hyperaccretion systems, which can launch relativistic jets as shown in Figure 1. If the jet can break out from the envelope and circumstellar medium, then a GRB will be triggered. Subsequently, the jet will be decelerated in the

media or winds to produce the multi-band GRB afterglows. Otherwise, the choked jets might produce luminous jet-driven supernovae (SNe).

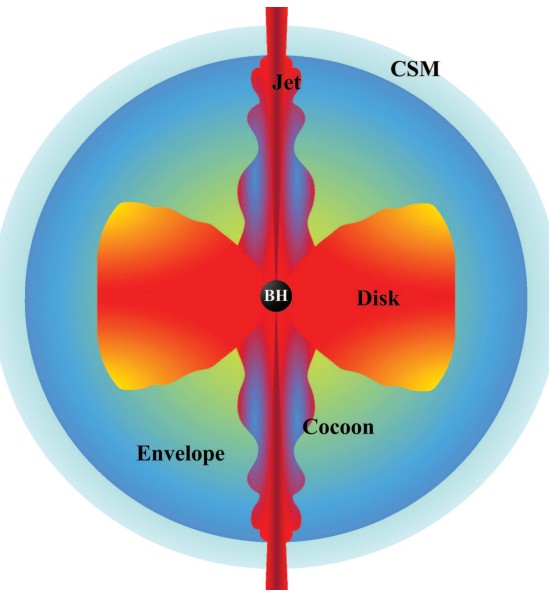

**Figure 1.** Schematic picture of a BH hyperaccretion in the center of a collapsar.

In the collapsar model, the greatest uncertainty is how the BH hyperaccretion system launches relativistic jets. So far, two well-known mechanisms have been proposed: neutrino annihilation and Blandford-Znajek (BZ) mechanism [13]. In the first case, if the accretion rate is very high ($\sim$0.001–10 $M_\odot$ s$^{-1}$), the inner region of the disk would be extremely hot and dense and photons are trapped in the disk. Neutrinos are emitted from the disk and annihilate in the space out of the disk to produce relativistic electron–positron jets. This accretion disk is called a neutrino-dominated accretion flow (NDAF), whose properties have been widely investigated [14–29]. For NDAFs in collapsars, the envelope would further collimate the jets. For the BZ mechanism, the BH rotation energy can be converted into the Poynting flux jet via a large-scale poloidal magnetic field threading the BH horizon to power GRBs [30–38]. Besides the collapsar model, the core collapse of massive star ($\sim$10–30 $M_\odot$) may also produce a millisecond magnetar (a rapidly spinning, supramassive, strongly magnetized NS [39–43]), which can launch relativistic jets and produce GRBs. LGRBs and their associated super-luminous SNe have been studied in the scenario of a magnetar produced by a massive star [44,45].

Another important physical process in the collapsar is jet propagation. Even if the central engine operates successfully, there is no guarantee that the jets can produce GRBs. After the jet is successfully launched, one of the main obstacles to producing GRBs is the star envelope that may prevent jet propagation. If the central engine turns off well before the jet head breaks out from the collapsar, the jet would become non-relativistic and thus incapable of creating a GRB. The observed GRB duration reflects only the duration of the central engine after the jet breaks out. Therefore, the properties of GRBs from collapsars are mainly determined by the jet propagation, and collapsars may produce failed GRBs. Besides, collapsars may produce a variety of outbursts, including X-ray flashes and jet-driven SNe. These phenomena are closely related to jet propagation.

Collapsars are also believed to be an important multimessenger sources to release neutrinos and gravitational waves (GWs). Multimessenger observations are essential to study the physics of collapsars, especially for the hidden BH hyperaccretion system. In fact, most observed electromagnetic signals from GRBs are produced in regions far away from the central engine. Only neutrinos and GWs can directly provide us with the information hidden deep inside the collapsar center. Besides, neutrinos and GWs can reach us without losing information of physical conditions near the BH hyperaccretion system. Moreover,

if collapsar fails to produce the GRB and SN, almost no electromagnetic signal can be detected. Then multimessenger signals are the only tool to investigate such events.

In this review, we summarize the theoretical progress on the BH hyperaccretion in the collapsar scenario. In Section 2, we introduce jet propagation in collapsars. We also discuss the feedback of disk outflows in collapsars. The signatures of the progenitors for producing LGRBs, SGRBs, and failed GRBs in the collapsar scenario are exhibited. In Section 3, the neutrino emission of NDAFs in the center of collapsars is presented. We discuss the effects of progenitor mass and metallicity on the neutrino spectra of NDAFs and compare the neutrino spectra of NDAFs with that of PNSs. The GWs from anisotropic neutrino emission of NDAFs in the center of collapsars are introduced in Section 4. We present the effects of progenitor mass and metallicity on the GW strains of NDAFs. We summarize the GW emission in the different phases of collapsars (the collapsar, central engine, and GRB jet phases). A brief summary is made in Section 5.

## 2. Jets and Outflows from BH Hyperaccretion in Collapsars

### 2.1. Jet Propagation

In past decades, jet propagation in collapsars has been investigated in both analytical [46–53] and numerical works [54–64]. Based on these investigations, we briefly review the general features of jet propagation in collapsars. After a cold relativistic jet is launched from the central engine, it pushes the collapsar matter leading to the formation of a shocked region at the jet head. The jet head is composed of two shocks. One is the forward shock, which sweeps the collapsar matter, and the other is the reverse shock, which decelerates the head of the jet. Both types of shocked matter are divided by the contact discontinuity. The velocity of the jet head can be calculated from the pressure balance at the contact discontinuity as [46,50,51,65]

$$\beta_{\rm h} = \frac{1}{1 + \tilde{L}^{-1/2}},$$

(1)

where

$$\tilde{L} \equiv \frac{L_j(t - r_{\rm h}/c)}{\pi r_{\rm h}^2 \theta_j^2 \rho(r_{\rm h}) c^3}$$

(2)

and $\theta_j$ is the jet half-opening angle and $L_{\rm j}$ is the jet luminosity. $r_{\rm h}(t) = \int_0^t c\beta_{\rm h} dt$ is the radius of the jet head.

If the velocity of the jet head is nonrelativistic ($\beta_{\rm h} < 1$), the shocked matter jet head is pushed sideways to form a cocoon structure around the jet [46]. Then the jet energy goes through the shocked region into the cocoon before the jet break out of the collapsar surface. If the jet head can break out from the collapsar and the velocity of the jet head is larger than that of the cocoon, it can contribute to the prompt high-energy emission and a GRB is triggered [46,66]. In the rest frame, the duration of GRB is $t_{\rm GRB} = t_{\rm eng} - t_b$, where $t_{\rm eng}$ is the duration of the central engine and $t_b$ is the jet breakout time. For a more detailed description of the jet-cocoon model, see Bromberg et al. (2011) [65].

### 2.2. Progenitor Stars

In the collapsar scenario, the progenitor stars of GRBs have been widely investigated. In the original proposal of Woosley (1993) [6], LGRBs are expected to come from the collapse of a Wolf-Rayet (WR) star. From the theoretical point of view, WR stars have no hydrogen and helium envelopes, and thus jets can penetrate them more easily. Further modeling of the collapsar model of LGRBs [8,54,55,67] shows that relativistic jets can break out from WR stars and produce GRBs. Matzner (2003) [46] constrained the progenitor of GRBs by modeling the dynamical interaction between a relativistic jet and a star envelope surrounding it. In this work, the life time of the central engine is assumed to be comparable to the observed duration of the prompt phase of GRBs. He concluded that only compact carbon–oxygen WR stars or helium post-WR stars can produce GRBs, while very massive stars such as Population III stars with massive envelopes are unlikely to be progenitors

of GRBs. So far, some LGRBs were associated with broad-line SNe Ib/c [5,68,69], which supported that GRBs come from carbon–oxygen WR stars.

Although WR stars are the most plausible LGRB progenitors, subsequent studies have shown that massive stars with supergiant hydrogen envelopes also can produce GRBs. Especially for some ultra-long GRBs (ULGRBs, [70,71]), they have ultra-long duration of the prompt emission with $\sim 10^4$ s, which cannot be explained by the anticipated central engine lifetimes of carbon–oxygen WR stars. In the collapsar scenario, the fallback accretion of a progenitor envelope [37,72] or direct envelope collapse of a more massive and extended progenitor [73] may cause ultra-long duration of the central engine. Suwa & Ioka (2001) [47] first predicted Population III stars also potentially create GRBs and the duration of GRBs can be very long. In their work, they employed three representative progenitors: Population III stars, WR stars, and red supergiant (RSG). They found that WR stars give rise to normal GRBs. Assuming the accretion-to-jet conversion efficiency is close to the normal GRBs, they calculated the jet propagation in very massive Population III stars. They analytically showed that Population III stars can produce GRBs even if they have supergiant hydrogen envelopes, thanks to the long-lasting powerful accretion of the envelope itself. The total energy injected by the jet is very large, however, more than half is hidden in the star. As for RSG, the jet head is slower than the cocoon and the later size of the cocoon becomes comparable to the radius of the jet head. Therefore, the jet propagation in RSGs gives rise to a spherical explosion but not a collimated GRB. Other works [50,56] also suggested that RSGs cannot be the progenitor of LGRBs in the collapsar scenario.

Nagakura et al. (2012) [63] investigated the propagation of jets in Population III stars by two-dimensional axisymmetric simulations. They adopted two kinds Population III stars: massive Population III stars and light Population III stars. Massive Population III stars are the first stars, which are supposed to be formed with a huge mass ($M > 100 \ M_\odot$) [74,75]. Light Population III stars are primordial but affected by radiation from other stars and are less massive ($M < 100 \ M_\odot$). Moreover, their relativistic hydrodynamic simulations first consider the negative feedback of the jet on the accretion. Their numerical calculations showed that the accretion-powered jet can potentially break out relativistically from the outer layers of Population III progenitors if the accretion-to-jet conversion efficiency is larger than a certain level. Otherwise, no explosion or some failed spherical explosions occur. They also verified that the central engine can last very long, >1000 s for massive Population III stars and >100 s for light Population III stars because of the accretion supply of the huge envelope.

Blue supergiants (BSGs) were also proposed as progenitors of GRBs in the collapsar scenario [73]. Nakauchi et al. (2013) [76] suggested that a metal-poor BSG collapsar can explain ULGRBs with superluminous-supernova-like (SLSN-like) bumps. In their model, the duration of ULGRBs can be explained by the accretion of the massive hydrogen envelopes of the BSG, while the SLSN-like bumps can be attributed to the so-called cocoon fireball photospheric emissions. Since a large cocoon is inevitably produced during the relativistic jet piercing through the BSG envelope, they suggested that this component can be smoking gun evidence of the BSG model for ULGRBs. If the observer locates along the off-axis direction from the ULGRB jet, only the SLSN-like component can be detected. They suggested that the GRB 111209A, a ULGRB with an SLSN-like bump, can be interpreted by the collapsar jet scenario of BSG progenitors. In principle, the most luminous GRBs are detectable up to redshift $z \sim 100$, while their afterglow are detectable up to $z \sim 30$. These signals are powerful probes of the high-$z$ universe. Nakachui et al. (2012) [56] suggested that GRBs from BSGs at $z \sim 9$ might be detected as long-duration X-ray-rich GRBs. For a higher redshift $z \sim 19$, GRBs from BSGs might be detected as long-duration X-ray flashes.

In Matsunmoto et al. (2015) [50], they considered supermassive Population III stars ($\sim 10^5 \ M_\odot$) as progenitor stars of collapsars. Supermassive Population III stars are larger in radius than Population III stars of mass 10–1000 $M_\odot$ or BSGs and have radii at least as large as RSGs. However, they found that jets are able to break out of the thick envelope of supermassive Population III stars. In contrast to RSGs, the envelopes of supermassive

Population III stars are dominated by pressure and have a steeper density gradient, which is benefit for jet propagation. They concluded that supermassive Population III stars forming in proto-galaxies can produce violent ULGRBs with a duration of $\sim 10^4$–$10^6$ s. Such GRBs are very energetic explosions and sweep up or blow off the matter in proto-galaxies.

### 2.3. Central Engines

The central engines of GRBs are expected as BH hyperaccretion systems in the collapsar scenario. There are mainly two candidates for jet production of the central engine: neutrino annihilation and the BZ mechanism. In general, the efficiency of the BZ mechanism is higher than that of neutrino annihilation. For BH hyperaccretion disks with the same BH spin parameter and accretion rate, the BZ luminosity is larger by about two orders of magnitude than neutrino annihilation luminosity [25,38]. If one considers that two mechanisms have the same conversion efficiency to power a certain GRB, the accretion rate or the BH spin parameter for the BZ mechanism would be lower than those for the neutrino annihilation.

Nagakura (2012) [63] investigated the propagation of neutrino-driven jets in WR stars in detail. They performed two-dimensional, relativistic hydrodynamical axisymmetric simulation of the accretion and subsequent jet propagation. They used the analytical formula proposed by Zalamea & Beloborodov (2011) [77] to estimate neutrino luminosity. The results showed that neutrino-driven jets in rapidly spinning WR stars can produce GRBs, while those propagating in slower rotating progenitors fail to break out due to insufficient kinetic power. Neutrino-driven jet propagation in slowly rotating WR stars may generate low-luminosity or failed GRBs. Besides, neutrino annihilation may not be suitable for Type II collapsar. MacFadyen et al. (2001) [9] studied the possible production of SNe and GRBs of Type II collapsar. They found that the typical accretion rate when most of the matter falls back in a Type II collapsar is 1–2 orders of magnitude less than in Type I collapsar. Therefore, the neutrino annihilation mechanism can not produce GRBs in Type II collapsar, while magnetohydrodynamic models may be suitable for Type II collapsar. Moreover, neutrino-driven jets are unlikely to break out from Population III stars. Suwa & Ioka (2001) [47] showed that neutrino annihilation is not effective for jets breaking out from the huge envelope of Population III stars, while the BZ mechanism works.

Overall, the BZ mechanism is more suitable for the collapsar model than neutrino annihilation, especially when accretion of central engine is not powerful enough, and ULGRBs are supposed to be related to the BZ mechanism. In addition, neutrino annihilation and the BZ mechanism may coexist in a BH hyperaccretion system [25,28,51]. For the BZ process, the required large-scale magnetic field may need to be maintained by the NDAF. The neutrino annihilation luminosity mainly contributes to the jet luminosity at the early stage of accretion. When the accretion rate decreases, the neutrino related process will be terminated and the BZ luminosity dominates the jet luminosity.

### 2.4. Disk Outflows

The disk outflows in collapsars are discussed in Liu et al. (2018) [51]. They introduced two kinds of outflows: outflow I and outflow II. Outflow I is launched when the matter of the envelope falls onto the outer boundary of the disk due to angular momentum redistribution. Outflow II is from the disk and will be strong when the mass accretion is high [51]. Then only a few percent of the supplied mass is eventually accreted into the BH. After considering the strong outflow from the disk, they found the inner accretion rate is lower than the ignition accretion rate of NDAFs; then, the BZ mechanism contributes to the jet luminosity. In their inflow-outflow model, they found BZ jets can break out from various types of progenitor stars and produce LGRBs and ULGRBs. Combining with GRB observations, they studied the masses and metallicities of the progenitors of LGRBs and ULGRBs. The results displayed that LGRBs lasting from several seconds to tens of seconds in the rest frame may originate from some zero-metallicity stars or solar-metallicity, massive ($M \geqslant 34 \, M_\odot$) stars. ULGRBs, such as GRB 111209A, can be produced by a fraction of low-

metallicity stars, including Population III stars. Then, Song & Liu (2019) [52] considered the SN ($^{56}$Ni) bumps in the inflow-outflow model. SN bumps are the late-time optical bumps in the afterglows of GRBs, which mainly originate from the decay of $^{56}$Ni. They assumed that the SN bump is powered purely by $^{56}$Ni synthesized in the outflows from the disk. GRB jets are powered by the BZ mechanism. As a result, there is competition between the luminosities of LGRBs and those of the corresponding $^{56}$Ni bumps because of the material distribution between the disk inflows and outflows. Comparing with GRB luminosity and $^{56}$Ni mass derived from the data of GRB-SN data, they constrained the feature of LGRBs and ULGRBs. They concluded that LGRBs-SNe can be produced by low-metallicity stars or massive solar-metallicity stars. If very strong outflows are launched from the disks, most of the massive low-metallicity stars could produce ULGRBs like GRB 111209A. Consequently, the outflows are potentially the main element factories.

Outflows from the disks will also generate feedback effects. Liu et al. (2019) [53] investigated the feedback mechanism of the outflows from the disks might exist in stellar scale collapsars. The outflows from the disk would be intercepted by the envelope, and there are interactions on the masses and energies between the outflows and progenitors in collapsars. Then materials may recycle via accretion by the BH, which subsequently prolongs the accretion timescale and fluctuates accretion rates. They found that this feedback mechanism of outflows can explain SN iPTF14hls. This SN is an unusually bright, long-lived SN (over 600 days), whose light curve has at least five peaks. In their feedback model, iPTF14hls might be a jet-driven SN. The jets cannot break out from the envelope or circumstellar medium, their energy is injected into the circumstances to power highly anisotropic explosions. The feedback of the strong outflow results in the unusual characteristics of this jet-driven SN. According to their estimations, they suggested that iPTF14hls may last no more than approximately 3000 days, and the luminosity may quickly decrease in the later stages.

*2.5. GRB Timescale*

Multimessenger observations indicated that SGRBs and LGRBs should be powered by ultrarelativistic jets launched from BH hyperaccretion in compact object mergers and massive collapsars, respectively. However, the duration is sometimes not a reliable indicator of the GRB physical origin. So far, some LGRBs have the statistical properties of SGRBs and are believed to come from compact object mergers [78,79], while some SGRBs are considered to be produced by massive collapsars [80–83]. Actually, there are two ways that collapsar may produce SGRBs. First, SGRBs from collapsars can be naturally caused by the "tip-of-iceberg" effect [84]. If the majority of gamma-ray emission episodes are below the detection threshold of GRB detectors, a real LGRB may be observed as a "short" one. Second, collapsars indeed produce SGRBs. In the collapsar model, the observed GRB duration reflects only the duration of the central engine after the jet breaks out. If the observed timescale is less than 2 s, this event may be classified as an SGRB. As a result, the duration of GRB mainly depends on jet propagation in the collapsar. Wei et al. (2022) [85] investigated the propagation of jets in collapsars with the core-envelope structure [86]. The density profiles can be divided into two parts: star core and envelope. The density profiles of the core and the envelope can be approximated as $\rho_{cor}(r) \propto r^{-k_1}$ and $\rho_{env}(r) \propto r^{-k_2}$, respectively. The boundary between the core and envelope is set as $r_1$. Then one can construct a series of density profiles by changing the values of $r_1$ and $k_2$. The structure of the core and envelope are determined by $k_1$ and $k_2$, respectively. The value of $k_1$ is set to 2.5, and the value of $k_2$ varies from 5 to 40.

In Wei et al. (2022) [85], they adopted two jet models for the jet-producing mechanisms. They assumed that jet is formed when the mass of BH reaches 3 $M_\odot$ because it has little effect on the luminosity [87] and set $t = 0$ at this time. In the first jet model, they assumed that the jet is driven by neutrino annihilation. Then the jet luminosity can be estimated as the neutrino annihilation luminosity: $L_j = L_{\nu\bar{\nu}}$. In the second model, they assumed that the jet is driven by the BZ process and neutrino annihilation together.

The GRB duration $t_{GRB}$ and isotropic energy of prompt emission $E_{\gamma,iso}$ of different collapsars for the neutrino-driven jet model are shown in Figure 2a,b, respectively. The mass supply to the BH hyperaccretion at the initial accretion stage is from the core, which becomes larger as $r_1$ becomes large. Therefore, a larger $r_1$ corresponds to a longer duration of the central engine. Besides, the accretion of the envelope may contribute to the duration of the central engine, especially when the envelope is thick. There is competition between mass supply onto the BH hyperaccretion and jet propagation into the envelope. For the collapsar with a thick envelope, the accretion of the envelope can enhance the mass supply onto the BH hyperaccretion and increase the duration of the central engine, while jets are difficult to break out from the collapsar. For the collapsar with a thin envelope, the duration of the central engine would be short, while jets can break out from the collapsar quickly.

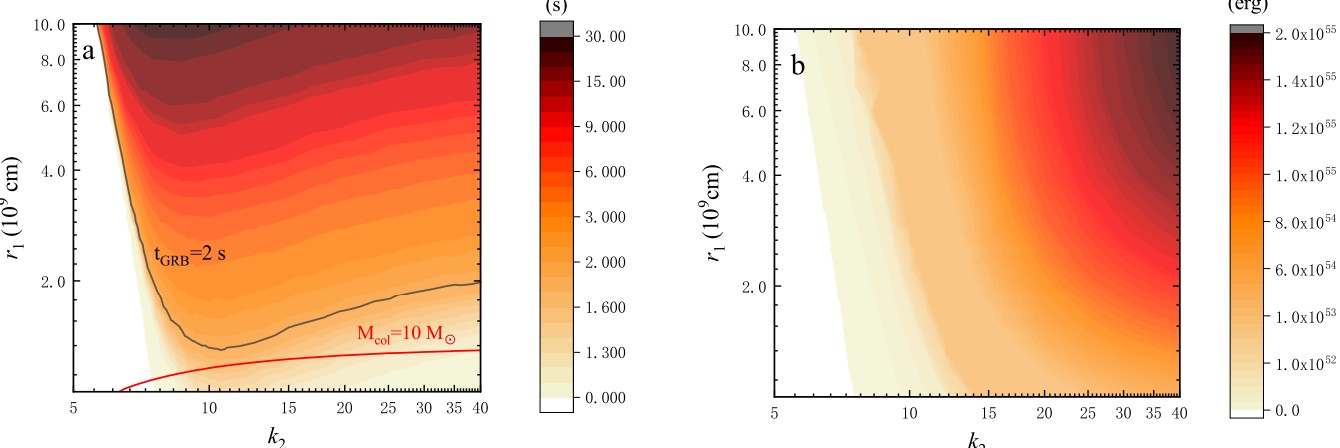

**Figure 2.** Panel (**a**): GRB duration $t_{GRB}$ as function of $r_1$ and $k_2$ for the neutrino-driven jet model. Black line corresponds to $t_{GRB} = 2$ s. Red line corresponds to collapsar mass $M_{col} = 10\,M_\odot$. Panel (**b**): isotropic energy $E_{\gamma,iso}$ as function of $r_1$ and $k_2$. (adapted with permission from Ref. [85]).

Figure 2a displays the parameter space for GRBs with different durations. The blank region corresponds to the case in which the jet is chocked in collapsar. The black line corresponds to $t_{GRB} = 2$ s. Above this line, collapsars produce LGRBs. A collapsar with a large core and a thin envelope is more likely to produce an LGRB. Below this line, collapsars would produce SGRBs or failed GRBs. Note that SGRBs can be produced regardless of the thickness of the envelope. For a collapsar with a thick envelope, an SGRB is mainly caused by the fact that the duration of the jet propagation is similar to the duration of the central engine. For a collapsar with a thin envelope, an SGRB is caused by the fact that the duration of the central engine itself is short. Figure 2b displays the $E_{\gamma,iso}$ of different collapsars for the neutrino-driven jet model. $E_{\gamma,iso}$ increases as $k_2$ increases. This is because that the jet takes a long time to break out of the collapsar and a large part of the jet energy is consumed in the envelope. Therefore, the duration and $E_{\gamma,iso}$ of GRBs can help constrain the density profiles of GRBs.

The GRB timescale $t_{GRB}$ for the BZ jet model is displayed in Figure 3a. Obviously, collapsars are more likely to produce LGRBs. SGRBs are produced only when $k_2$ is large. Actually, the jet can easily break out of the collapsars for the BZ jet model even if the envelope is thick. The result shows that the jet is nonrelativistic when $k_2 < 5$. Figure 3b displays the $E_{\gamma,iso}$ of different collapsars for the BZ jet model. Similarly, $E_{\gamma,iso}$ increases as $k_2$ increases. For the same density profiles, $E_{\gamma,iso}$ in Figure 3b is large than that in Figure 2b. Overall, collapsars with small $k_2$ can produce GRBs for both jet models, which support the results of previous works [56,63] that light Pop III stars may produce GRBs.

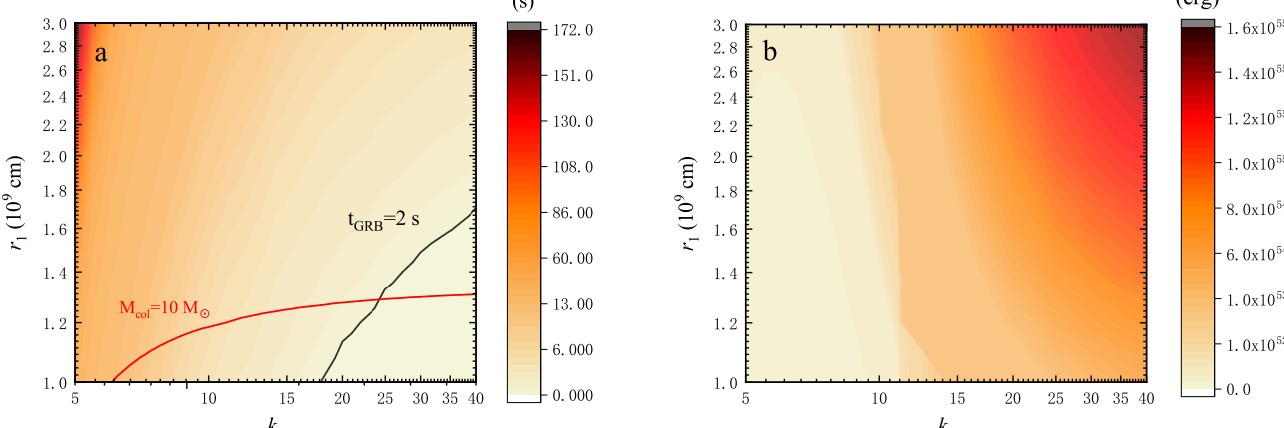

**Figure 3.** Panel (**a**): GRB duration $t_{GRB}$ as function of $r_1$ and $k_2$ for the BZ jet model. Black line corresponds to $t_{GRB} = 2$ s. Red line corresponds to collapsar mass $M_{col} = 10\,M_\odot$. Panel (**b**): isotropic energy $E_{\gamma,iso}$ as function of $r_1$ and $k_2$. (adapted with permission from Ref. [85]).

As a result, collapsar can produce LGRBs, SGRBs, and failed GRBs for both jet models. The duration and $E_{\gamma,iso}$ of GRBs from collapsars are determined by the mass supply and jet propagation together. Generally, a massive collapsar with a thin envelope is more likely to produce LGRBs. For the neutrino-driven jet model, both thick and thin envelopes can result in the production of SGRBs. For the BZ mechanism, jets can easily break out of collapsars, and only collapsars with thin envelopes can give rise to SGRBs. Duration and isotropic energy $E_{\gamma,iso}$ of GRBs can help constrain the density profiles of collapsars.

## 3. MeV Neutrinos from NDAFs in Collapsars

The inner region of NDAFs is extremely hot and dense, and photons are completely trapped. A large number of neutrinos escape from the surface of the disk to carry away the viscously dissipated BH gravitational energy. An NDAF is ignited when the ignition radius $r_{ign}$ emerges in the disk. Here, $r_{ign}$ is defined as the radius satisfied with $Q_\nu^-/Q_{vis} = 1/2$, where $Q_\nu^-$ and $Q_{vis}$ are the neutrino cooling rate and the viscous heating rate, respectively [19,77,88]. The corresponding mass accretion rate of the disk is $\dot{M}_{ign}$, which is mainly related to the viscous parameter of the disk and the BH spin. In the region of $r < r_{ign}$, the neutrino emission switches on and neutrino cooling dominates. In the collapsar scenario, the amount of mass accreted onto the BH decreases with time [12,51]. For a rapidly rotating BH with $3\,M_\odot$ surrounded by an NDAF with low viscosity, $\dot{M}_{ign}$ is about $0.001\,M_\odot\,\mathrm{s}^{-1}$ [28]. When $\dot{M} < \dot{M}_{ign} \sim 0.001\,M_\odot\,\mathrm{s}^{-1}$, neutrino emission can be ignored.

There are many neutrino cooling processes in the disk, including the Urca processes, electron–positron pair annihilation, plasma decay, and nucleon-nucleon bremsstrahlung. The dominant neutrino cooling process is the Urca process, and electron neutrinos and antineutrinos are the dominant neutrino flavors. A small fraction of neutrinos and antineutrinos escape from the disk surface and annihilate above the disk, which would produce a thermally dominated fireball to power GRBs. This process only consumes about 1% of the total neutrino emission energy [14,21,24]. Thus, annihilation effects can be neglected when one calculates the neutrino emission of the NDAFs.

The neutrino emission from NDAFs has been studied by many previous works. The detectability of a nominal NDAF by Super-K has been investigated [89–91]. Liu et al. (2016) [92] calculated the electron neutrino and antineutrino spectra of NDAFs by fully taking into account the general relativistic effects. The neutrino luminosity of a typical NDAF can reach $10^{50}$–$10^{52}$ erg s$^{-1}$ peaking at ~10–20 MeV, and NDAFs are expected to be detected by many MeV neutrino detectors. Wei et al. (2019) [93] studied the neutrino emission from NDAFs in the collapsar scenario and the effects of the properties of progenitor

stars on the neutrino spectra. The mass of a collapsar has little influence on the neutrino spectrum, and a low metallicity is beneficial to the production of low-energy neutrinos.

In the following, we introduce the MeV neutrino emission from NDAFs in the center of collapsars.

### 3.1. Neutrino Spectra of NDAFs

In Wei et al. (2019) [93], they adopted the pre-SN model [94–96] as progenitor model to study the effects of mass and metallicity on the multimessenger emission of NDAFs. These progenitor models were evolved using KEPLER code [94,97] through all stable stages of nuclear burning until their iron cores became unstable and collapsible.

Considering the detailed neutrino physics, chemical potential equilibrium, neutrino trapping and nucleosynthesis, Xue et al. (2013) [24] investigated one-dimensional global solutions of NDAFs in Kerr metric. Based on the results, Liu et al. (2016) [92] derived the fitting formulae for the mean cooling rate due to electron neutrino losses, $Q_{\nu_e}$, in units of erg cm$^{-2}$ s$^{-1}$, and the temperature of the disk $T$, in units of K, as a function of the mean BH spin parameter, the accretion rate, and the radius ($M_{BH} = 3\,M_\odot$ adopted), i.e.,

$$\log Q_{\nu_e} = 39.78 + 0.15a_* + 1.19\log\dot{m} - 3.46\log r, \tag{3}$$

and

$$\log T = 11.09 + 0.10a_* + 0.20\log\dot{m} - 0.59\log r, \tag{4}$$

where $a_*(0 \leq a_* \leq 1)$ is the dimensionless BH spin parameter, and $\dot{m} = \dot{M}/M_\odot$ s$^{-1}$ and $r = R/R_g$ are the dimensionless mass accretion rate and radius, respectively. $R_g = 2GM_{BH}/c^2$ is the Schwarzschild radius.

The neutrino-cooling rate decreases with radius due to the drop in temperature and density. Therefore, neutrinos are mainly emitted from the inner region of the disk. As a result, the observed neutrino spectra are affected by general relativistic. Since the rest mass of neutrinos is much less than their kinetic energy, one can calculate the neutrino propagation in a manner similar to photon propagation near an accreting BH [98,99]. In Wei et al. (2019) [93], they used the ray-tracing method to calculate the neutrino propagation effects [100,101].

The time-integrated electron spectra of NDAFs in the center of collapsars are shown in Figure 4. The blue, red, green, and black curves correspond to metallicities of $Z/Z_\odot = 1$, 0.1, 0.01, and $10^{-4}$, respectively. Here, $Z_\odot$ is the metallicity of the Sun First, the total mass of the progenitor has little effect on the neutrino spectra. Second, the peak energies of the calculated spectra are approximately 10–20 MeV. For NDAF models, most high-energy neutrinos are emitted in the hyperaccretion stage with a high accretion rate. In the low energy range of the spectra, the amplitudes of the spectral lines increase with decreasing metallicity. This is because a lower metallicity corresponds to neutrino emission with a longer timescale. The low-metallicity star has a large envelope and the accretion of mass from envelope would contribute to the duration of NDAF. In the late accretion stage of an NDAF, the disk mainly emits low-energy neutrinos, which increases the amplitudes of the spectral lines of a progenitor with low metallicity. In the high energy range of the spectra, the amplitudes of the spectral lines depend on the initial mass accretion rate of the NDAF, which are determined by the metallicity and mass of the progenitor stars.

Note that the neutrino oscillations both inside the collapsar and in vacuum are not considered in our calculations. These effects would affect the shape of the neutrino spectra of NDAFs in the center of collapsars [102]. Neutrinos emitted from the central engine would pass through the envelope of the collapsar and undergo flavor transformation. In the envelope, neutrino flavor transformations have a more significant influence on the flux of electron neutrinos than on the flux of electron antineutrinos [103,104]. In a vacuum, however, neutrino flavor transformations have similar influences on the fluxes of electron neutrinos and electron antineutrinos. Considering the similar physical conditions and flavor distributions of NDAFs with SNe, neutrino oscillations may change the

neutrino spectra moderately and reduce the neutrino flux of NDAFs by at most a factor of 2–3 [92,103–105]. According to Kotake et al. (2006) [106], neutrino oscillations can be addressed in postprocessing to calculate the final neutrino signal that reaches detectors on Earth.

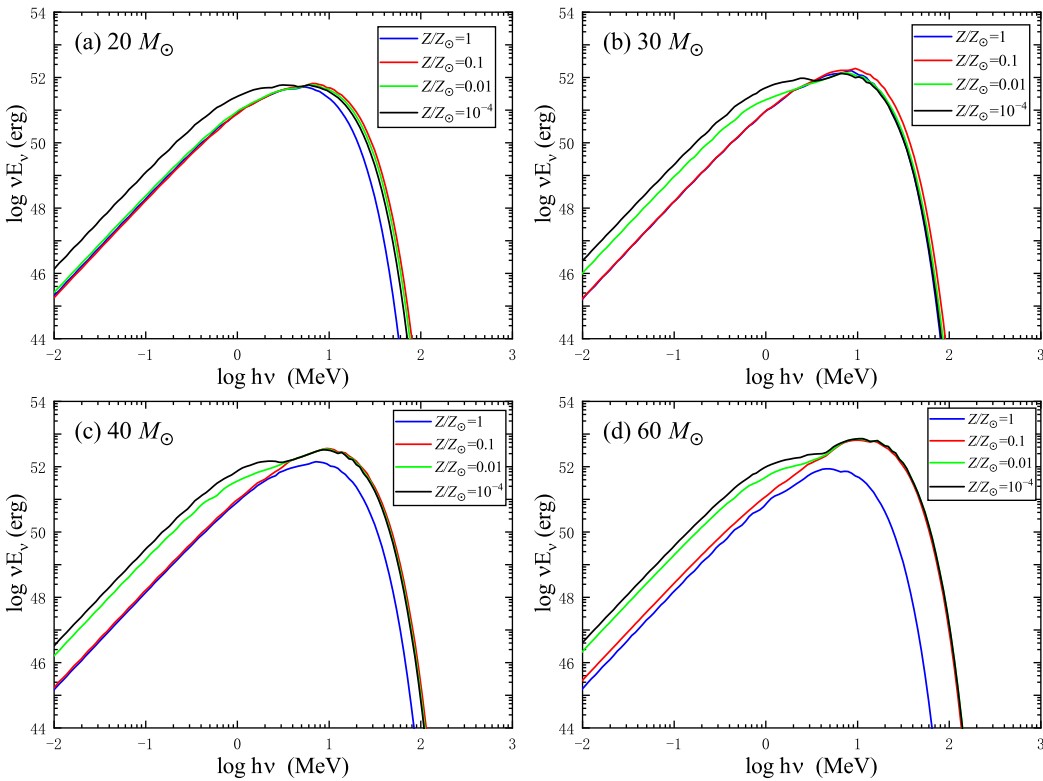

**Figure 4.** Electron neutrino spectra of NDAFs with different masses and metallicities. Panels (**a**–**d**) correspond to progenitor star masses of $M_{pro}/M_\odot$ = 20, 30, 40, and 60, respectively. The blue, red, green, and black curves correspond to progenitor star metallicities of $Z/Z_\odot$ = 1, 0.1, 0.01, and $10^{-4}$, respectively. (adapted with permission from Ref. [93]).

*3.2. Comparisons with PNSs*

Besides BHs, NSs could also be the compact remnants of the collapsars [107]. PNS is a newborn NS, which is hot, proton-rich, and contains a large number of neutrinos and degenerate electrons [108]. Neutrino emission plays an important role in deleptonization and cooling of a hot PNS during the Kelvin-Helmholtz epoch. Generally, it takes a hot PNS tens of seconds to cool to form a cold and deleptonized NS [109]. Although neutrinos are nearly massless particles and are only affected by the weak interaction with extremely small scattering cross-sections, the matter of a PNS is quite opaque at extremely high temperature and density. Then neutrinos are trapped in PNS and escape by diffusion [110,111].

Neutral current scattering and charged current absorption reactions are the main mechanisms contributing to the opacity [110–112]. Opaque neutrino emission from PNSs has the form of blackbody emission regardless of the mechanism [113,114]. The neutrino spectrum of PNS conforms to the Fermi-Dirac energy distribution, which is determined by the neutrinospheric temperature.

The cooling of PNSs has been investigated by many numerical works [109,110,115]. Wei et al. (2019) [93] adopted part of the simulation results of Pons et al. (1999) [109] as typical solutions to describe the time-integrated electron neutrino spectrum of a PNS, which is shown in Figure 5. For comparison, neutrino spectra of NDAFs corresponding to progenitor stars with ($M_{pro}/M_\odot$, $Z/Z_\odot$) = (20, 0.01) and (40, 0.01) are also shown in Figure 5. In the low-energy region of spectra, the differences are obvious. The spectra of the NDAFs definitely exhibit a more gentle ascent than the PNS spectrum. Above several

MeV, the amplitudes of the NDAF spectra are about one order of magnitude lower than those of the PNS spectra.

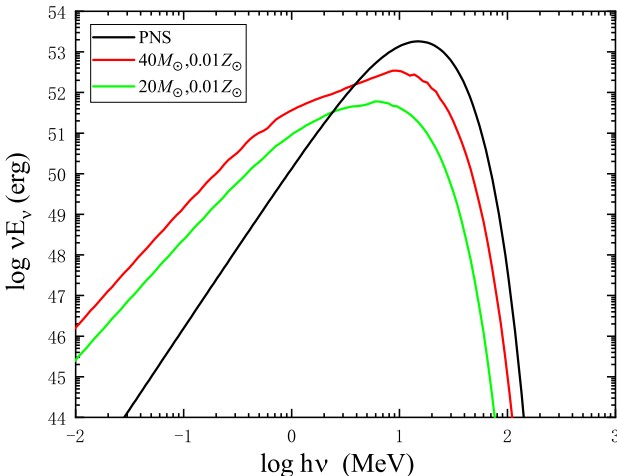

**Figure 5.** Time-integrated electron neutrino spectra of a PNS and NDAFs in the center of a collapsar. The black curve corresponds to a PNS, and the green and red lines correspond to NDAFs in the center of progenitor stars with $(M_{\mathrm{pro}}/M_\odot, Z/Z_\odot)$ = (20, 0.01) and (40, 0.01), respectively. (adapted with permission from Ref. [93]).

According to some core-collapse simulations, a fraction of the total emitted neutrinos are emitted in a hyperaccretion phase lasting about several hundred milliseconds before PNS cooling [115,116]. However, most of the neutrinos are emitted during the subsequent tens of seconds of PNS cooling and deleptonization [110]. Generally, the energies of neutrinos emitted during the hyperaccretion phase are higher than that during the PNS cooling phase. Therefore, a high-energy tail will be superimposed on the spectrum [116]. Nonetheless, the neutrino energies from the PNS are still higher than those of an NDAF in the high-energy band and can be distinguished.

In addition, for Type II collapsar, the central core forms an NS first and then collapses to a BH surrounded by an accretion disk due to the fallback process. In this case, the neutrinos in collapsars are powered by two sources, i.e., the PNS and NDAF. Since the timescale over which an NS collapses to a BH is unknown, one can use a neutrino light curve to examine this process. The neutrino luminosity of a PNS is higher than that of an NDAF and therefore the neutrino flux would show a rapid decrease when a PNS collapses to a BH [92,117].

*3.3. Detection*

The detection of MeV neutrinos from SN1978A in the Large Magellanic Cloud (LMC) [118–120] initiated a new era of neutrino astronomy.

The capability for worldwide MeV neutrino detection has improved quickly. For an SN with total neutrino energy of $\sim 3 \times 10^{53}$ ergs at 10 kpc, about 170,000–260,000 neutrino events would be detected by the future Hyper-Kamiokande (Hyper-K) detector [121]. If such an event occurs in the LMC where SN1987A is located, about 7000–10,000 neutrino events are expected. Moreover, other projects on MeV neutrino detection, such as Super-K, Jiangmen Underground Neutrino Observatory (JUNO), and Deep Underground Neutrino Experiment, declared that they can also detect neutrinos with energies of many thousands of MeV from a nearby core-collapsar event [122]. The future liquid-scintillator detector LENA (Low Energy Neutrino Astronomy) will have a good performance in the low-energy energy band ($\lesssim$1 MeV) [123]. For the Galactic NDAFs and PNSs, LENA has the ability to detect a large number of sub-MeV neutrinos.

The detection rate of NDAFs has been investigated in Liu et al. (2016) [92]. If one takes the event rate of the SN Ib/c [124,125] as an optimistic event rate for NDAFs, the

expected detection rate for NDAFs in the Local Group is 1–3 per century for Hyper-K [92]. At this distance, however, only several neutrinos can be detected by Hyper-K, which is not useful for studying the properties of NDAFs. As a result, one should focus on the closer collapsar events and need more powerful detectors. According to the distribution of possible massive progenitor stars in the Milky Way, the most likely distance of the next collapsar event from the Sun is in the range of 12–15 kpc in the Galactic plane [126–128]. At such a distance, one might detect ∼1000 neutrino events from the central engine by LENA. Moreover, one may roughly distinguish the nature of the remnants of the core collapse of massive stars.

Due to the limitation of event rate of collapsar and detector, it is very hard to achieve direct detection of the MeV neutrinos from the central engine of nearby collapsar now. The joint multimessenger observations of the electromagnetic counterparts and GWs of the central engine are more likely to constrain the nature of the remnants. One can expect that future neutrino detectors with a low-energy threshold, high signal rate, and good energy resolution provide the high-statistics light curve and spectrum of a neutrino burst from the core collapse of a massive star. As a result, explosion mechanisms and neutrino physics, such as mass and its hierarchy, mixing, and oscillation, would be intensively studied [128].

## 4. GWs

The detection of a GW event from a binary NS merger system GW170817 [129] that was associated with electromagnetic signals marked that we have entered an era of multi-messenger astronomy. In the future, astrophysical sources including massive star collapse, BH hyperaccretion, rapidly rotating NSs, and other violent events in the universe might be detected by GW detectors. Moreover, such events are promising multimessenger transient sources, especially for massive star collapse.

GWs from GRB central engines have been studied by some previous works. Sun et al. (2012) [130] studied the GWs of jet precession based on NDAFs around BHs. They found that the jet and the inner part of the disk may precess along with the BH, which is driven by the outer part of the disk. This BH-disk precession system can emit GWs, which may be detected by Decihertz Interferometer Gravitational Wave Observatory (DECIGO)/Big Bang Observer (BBO) in the Local Group (<1 Mpc). GWs arising from anisotropic neutrino emission from NDAFs have been investigated [131–136]. Suwa et al. (2009) [131] first calculated GW signals produced by neutrinos from NDAFs around BHs. They considered that neutrino-induced GWs are detectable for ∼10 Mpc by DECIGO/BBO. Kotake et al. (2012) [132] studied GWs generated by asphericities in neutrinos emission of NDAFs in the context of the collapsar model by performing two-dimensional relativistic magnetohy-drodynamic simulations. They found that the GW amplitudes from anisotropic neutrino emission show a monotonic increase with time. Liu et al. (2017) [133] examined the BH spin and accretion rate impacts on the GW strains from NDAFs. They proposed that GWs from NDAFs are expected to be detected at a distance of ∼100 kpc/∼1 Mpc by the advanced LIGO/Einstein Telescope (ET) with a typical frequency of ∼10–100 Hz, and they compared GWs from different central engines of GRBs: NDAFs, BZ mechanism (no GW emission), and millisecond magnetars. For a certain GRB, the possible detected distance from NDAFs is about two orders of magnitude lower than that from magnetars, but at least two orders of magnitude higher than that from the BZ mechanism associated with weak neutrino annihilation. Moreover, the typical GW frequency for NDAFs is the same as that of the BZ mechanism associated with weak neutrino annihilation, ∼10–100 Hz, while the typical frequency for magnetars is ∼2000 Hz. Thus, the GWs released by the central engines of adjacent GRBs might help determine whether there is an NDAF, BZ jets, or a magnetar. Song et al. (2020) [134] calculated neutrinos and GWs from magnetized NDAFs with magnetic coupling (MC). They studied the structure, luminosity, MeV neutrinos, and GWs of magnetized NDAFs under the assumption that both the BZ and MC mechanisms are present. The typical neutrino luminosity of magnetized NDAFs is higher than that of NDAFs. If the magnetic coupling is dominant, the GW strains from magnetized NDAFs

will be stronger than those of NDAFs. In Wei et al. (2020) [135], they investigated the GW emission generated by the anisotropic neutrino emission from NDAFs in collapsar scenarios and the effects of the masses and metallicities of progenitor stars on the GW strains from NDAFs. Moreover, they summarized the GW emission in the different phases of collapsars. The primary detectable frequencies and strains in the three phases (the collapsar, central engine, and GRB jet phases) are different. As a result, GWs from collapsars can help constrain the characteristics of collapsars and central BH accretion systems.

In the following, we introduce GWs from collapsars.

### 4.1. GWs from NDAFs in Collapsars

The GW emission from a point source due to the anisotropic neutrino radiation was first proposed by Epstein (1978) [137]. Then many simulations investigated the neutrino-induced GWs from core-collapse SNe (CCSNe) [106,138–140]. Based on the methods applied to CCSNe, Suwa et al. (2009) [131] first derived useful formulas of the GW amplitude for axisymmetric emission of neutrinos from NDAFs. Assuming the emission of neutrinos is isotropic at any point of the disk surface, the nonvanishing GW amplitude can be given as [135]

$$h_+(t,\vartheta) = \frac{1 + 2\cos\vartheta}{3}\tan^2\left(\frac{\vartheta}{2}\right)\frac{2G}{Dc^4} \times \int_{-\infty}^{t-D/c} L_\nu(t')dt', \tag{5}$$

where $\vartheta$ is the viewing angle, $D$ is the distance from the observer to the source, and $L_\nu(t')dt'$ is the neutrino luminosity of the source. One can see the dependence of the GW amplitude on the viewing angle. $\vartheta = \pi/2$ corresponds to the case that the observer is located in equatorial plane of the disk, and the GW amplitude is the largest. The GW vanishes when the observer is located in the pole direction ($\vartheta = 0$).

The neutrino luminosity of the source mainly depends on the mass accretion rate of NDAFs. In the collapsar scenarios, the mass accretion rate onto the BH decreases over time. Xue et al. (2013) [24] calculated one-dimensional global solutions of NDAFs in the Kerr metric. According to the results, they fitted time-independent analytical formulas, and the neutrino luminosity $L_\nu$ is given by

$$\log L_\nu(\mathrm{erg\,s}^{-1}) \approx 52.5 + 1.17a_* + 1.17\log\dot{m}. \tag{6}$$

Here the spin of the BH is adopted as $a_* = 0.9$. The GWs from NDAFs depend on the neutrino luminosity, and the typical frequency is determined by the variabilities and duration of neutrino emission.

In order to get a GW spectrum, $L_\nu(t)$ is written in terms of the inverse Fourier transform as

$$L_\nu(t) = \int_{-\infty}^{+\infty} \tilde{L}_\nu(f)e^{-2\pi i f t}df \tag{7}$$

where $f$ is the frequency.

The local energy flux of GWs can be written as [131]

$$\frac{dE_{\mathrm{GW}}}{D^2 d\Omega dt} = \frac{c^3}{16\pi G}\left|\frac{d}{dt}h_+(t,\vartheta)\right|^2, \tag{8}$$

where $\Omega$ is the solid angle in the observer coordinate frame. Integrating over a sphere surrounding the source, the total energy emitted by GW can be obtained as

$$E_{\mathrm{GW}} = \frac{\beta G}{9c^5}\int_{-\infty}^{\infty} dt L_\nu(t)^2, \tag{9}$$

where $\beta \sim 0.47039$.

Then the GW energy spectrum can be deduced as

$$\frac{dE_{\mathrm{GW}}(f)}{df} = \frac{2\beta G}{9c^5}\left|\tilde{L}_\nu(f)\right|^2.$$

(10)

For a given frequency $f$, the characteristic GW strain is expressed as [141]

$$h_{\mathrm{c}}(f) = \frac{1}{R}\sqrt{\frac{2}{\pi^2}\frac{G}{c^2}\frac{dE_{\mathrm{GW}}(f)}{df}}$$

(11)

After one obtains the characteristic strain, the signal-to-noise ratios (S/Ns) obtained from the matched filtering for the GW detectors can be calculated. For an optimally oriented source, the S/N is given by

$$\mathrm{S/N}^2 = \int_0^\infty d(\ln f)\frac{h_{\mathrm{c}}(f)^2}{h_{\mathrm{n}}(f)^2},$$

(12)

where $h_{\mathrm{n}}f = \sqrt{5fS_{\mathrm{h}}(f)}$ is the noise amplitude, and $S_{\mathrm{h}}(f)$ is the power spectral density of the strain noise in the detector at frequency $f$.

The characteristic strains of the GWs from NDAFs at a distance of 10 kpc are shown in Figure 6. The blue, red, green, and black curves correspond to the progenitor masses of $M_{\mathrm{pro}}/M_\odot$ = 20, 30, 40, and 60, respectively. The gray lines represent the sensitivity curves (the noise amplitudes $h_{\mathrm{n}}$) of aLIGO, ET, Cosmic Explorer (CE), Laser Interferometer Space Antenna (LISA), Taiji, TianQin, DECIGO/BBO, and ultimate-DECIGO, respectively. The GWs from NDAFs in the center of collapsars might be detected by DECIGO/BBO and ultimate-DECIGO in the detectable frequency ∼1–10 Hz, and by aLIGO, CE, and ET in the detectable frequency ∼10–100 Hz at a distance of ∼10 kpc.

The mass accretion rate onto the BH in the initial accretion phase tends to slightly rise with the increase of the progenitor mass, the GW strains increase slightly. For the same progenitor mass, the accretion rate in the initial accretion stage shows little difference for the different metallicities. Although the metallicity can affect the duration of the NDAF in the collapsar, the neutrino cooling is invalid in the late accretion stage and GW emission mainly occurs in the early stage of NDAFs. Therefore, the progenitor metallicities have little effect on the GW strains of NDAFs.

*4.2. GWs from Collapsars*

Strong GWs are expected to be emitted during a gravitational collapse/explosion and by the resulting compact remnant [142]. The GW signals from massive collapsars, especially for Type II collapsars, are similar to those from normal CCSNe, whose GW emission have been widely studied [143,144]. If the collapsars or the resulting SN explosions are nonspherical such that the third time derivative of the quadrupole moment of the mass–energy distribution is nonzero, part of the liberated gravitational binding energy will be emitted in the form of GWs. Such nonsphericities may be caused by many effects such as rotation, convection, fragmentation instability, and anisotropic neutrino emission. These effects lead either to large-scale asphericities or small-scale statistical mass-energy fluctuations [137–139,145–154].

In the collapsar scenarios, a massive star may go through the collapse, bounce, and postbounce phase, then BH formation, the hyperaccretion phase, and the GRB jet phase [132,155]. Generally, one can divide this evolutionary process into three periods: the collapsar phase (from collapse and bounce to BH formation), central engine phase (hyperaccretion phase) and GRB jet phase. Type II collapsars are expected to go through the above three phases, while Type I and III collapsars may only go through central engine phase and GRB jet phase. The GW signals from these three different phases are displayed in Figure 7. The blue, purple, and orange shaded regions correspond to the collapsar phase,

NDAFs and GRB jet phase, respectively. In the following, we introduce the GW emission from different phases.

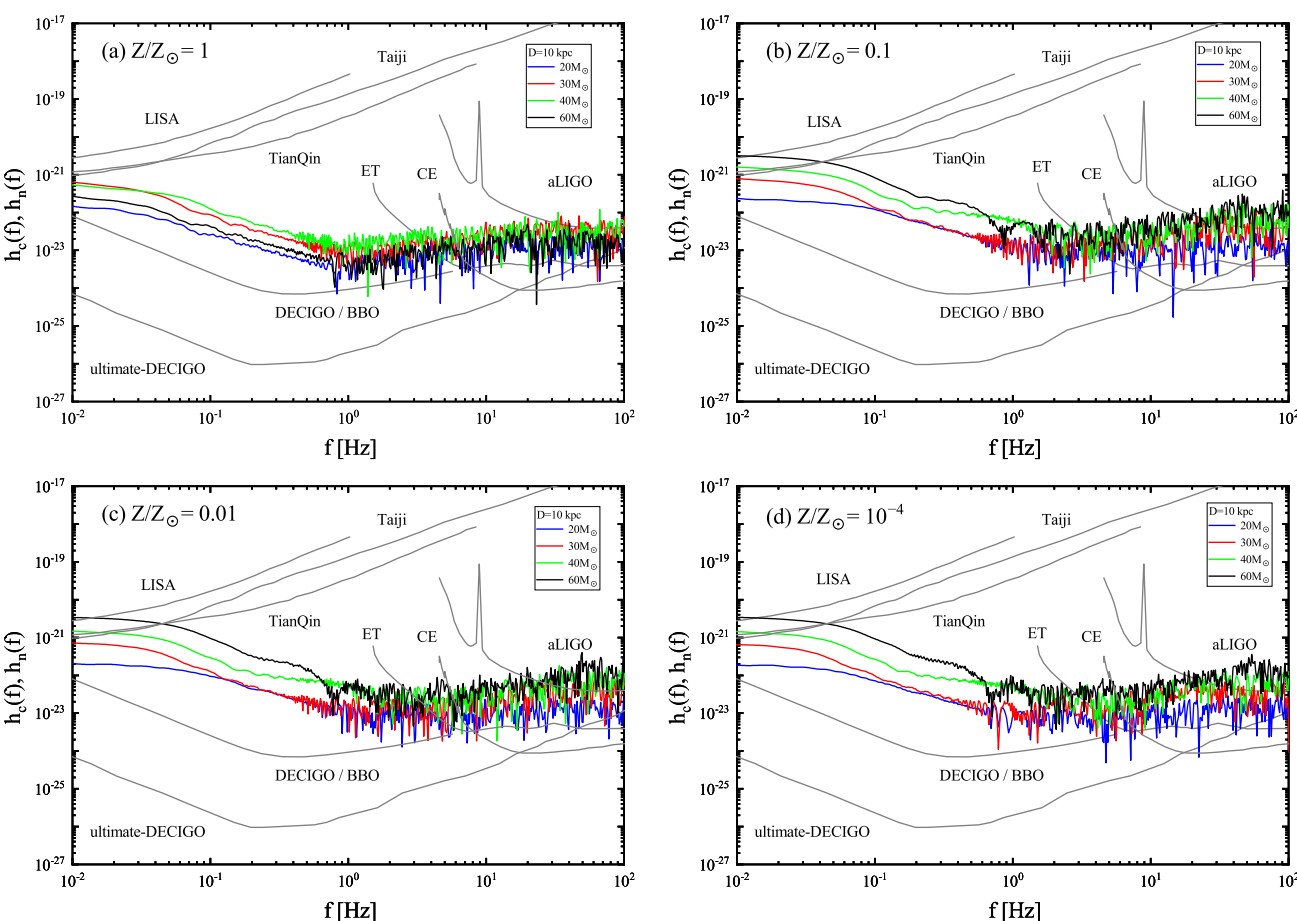

**Figure 6.** The strains of the GWs from NDAFs in the center of collapsars at the distance of 10 kpc. Panels (**a**–**d**) correspond to progenitor metallicities of $Z/Z_\odot = 1$, 0.1, 0.01, and $10^{-4}$, respectively. The blue, red, green, and black curves correspond to progenitor masses of $M_{\mathrm{pro}}/M_\odot = 20$, 30, 40, and 60, respectively. In all four figures, the gray lines show the sensitivity lines (the noise amplitudes $h_{\mathrm{n}}$) of aLIGO, ET, CE, LISA, Taiji, TianQin, DECIGO/BBO, and ultimate-DECIGO. (adapted with permission from Ref. [135]).

In the collapsar phase, rotating collapse and core bounce are the main GW source. Many original studies focused on GW emission during the collapse and bounce phase due to the star's changing quadrupole moment. A rough description of the possible evolution of the quadrupole moment is given in Fryer & New (2011) [142]. During the bounce phase, the shape of the core, the depth of the bounce, the bounce timescale, and the rotational energy of the core all would affect the GW signals. Generally, the typical frequency is expected to be 100–1000 Hz and the peak amplitude of GW is roughly proportional to the collapsar spin [106,143,144].

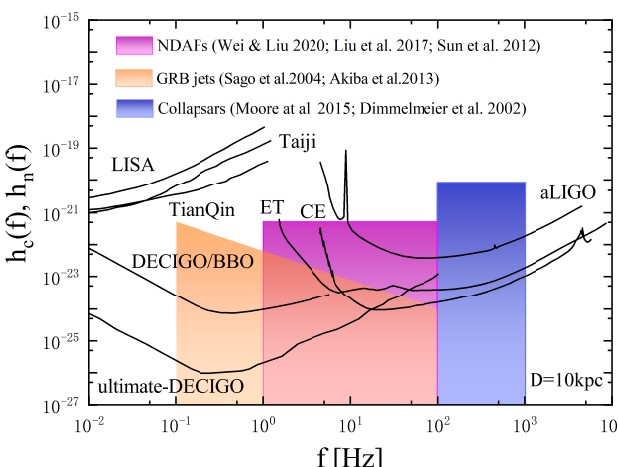

**Figure 7.** The characteristic amplitude of GWs from different sources in a collapsar. The blue, purple, and orange shaded boxes represent the collapsar phase [152,156], NDAFs [130,133,135] and GRB jet phase [157,158], respectively. The gray lines show the sensitivity lines (the noise amplitudes $h_n$) of aLIGO, ET, CE, LISA, Taiji, TianQin, DECIGO/BBO, and ultimate-DECIGO. The distance is 10 kpc. (adapted with permission from Ref. [135]).

In the postbounce phase, anisotropic matter motions associated with convection, anisotropic neutrino emission, and standing-accretion-shock instability (SASI, [140,159–163]) are the primary GW sources. Convection is a central feature of the postbounce evolution of massive stars [108,160,164–166]. After the bounce shock is formed, entropy- and lepton-gradient may drive prompt convection. Then, inside the PNS, the negative lepton gradient may drive PNS convection. In the post-shock heating region, the neutrino-driven convection may develop [167–169]. The SASI is expected to be caused by either a purely acoustic or an advective acoustic feedback cycle, resulting in the growth of perturbations in the stalled shock. After the SASI grows to nonlinear amplitudes, it would heavily distort the postshock region and affect convection. Both the SASI and convection are intrinsically multidimensional phenomena and produce large-scale accelerated mass motions, which would emit GWs. Ott (2009) [143] gave a semiquantitative summary of the GW emission by the aspherical fluid motions associated with the convection and SASI. Based on numerous previous numerical works [154,170,171], they concluded that the typical GW frequency $f$ is in the range of 100–1000 Hz and gave estimations of the typical GW strains at 10 kpc.

In the context of the core collapse of massive stars, the global asymmetries in the (precollapse) matter distribution [138,172], the convective overturn and SASI [140,170,171], and the rotationally deformed PNSs [106,154] may produce anisotropic neutrino emission. The neutrino-induced GW signals are slightly different from GW signals from matter motions. The GW waveforms from anisotropic neutrino emission have much less time variation structure, hence anisotropic neutrino emission produces stronger signals at low frequencies (below ∼100 Hz) [106,138,154]. Besides, the precollapse density inhomogeneities [138,139,172], nonaxisymmetric rotational instabilities [150,173], $g$-mode [170] and $r$-mode pulsations of PNSs [174], and aspherical mass ejection may contribute to the overall GW signature. Moreover, various physical mechanisms can also produce post-bounce asphericites, including the precollapse density inhomogeneities [138,139,172], nonaxisymmetric rotational instabilities [150,173], $g$-mode [170] and $r$-mode pulsations of PNSs [174], and aspherical mass ejection, which may contribute to the overall GW signature.

Many numerical works have investigated GW emission at the moment of the BH formation [155,175–177]. When a nonspherical PNS collapse to a BH, the GW emission may be caused by the rapidly shrinking mass-quadrupole moment of the PNS. At the initial BH formation and the subsequent accretion, the infalling matter may increase the mass and spin of the BH, and perturb the geometry of the BH, distorting it from the Kerr solution. This distortion may drive the nascent BH to ring in distinct harmonics, which

would emit GWs. The peak frequency from *g*-mode PNS oscillations at BH formation is expected to be above 2 kHz, while the GW emission from BH formation itself peaks in the kHz range [155,177]. However, those GW signals are very close to the limit of the current detectors and are difficult to detect. As a result, referring to the GW emission mechanisms and current GW detectors, the most promising detectable frequency is at 100–1000 Hz in the collapsar phase. For a fast rotating massive star, the average maximum amplitude of GWs at the distance *D* is estimated as [152,156]

$$h_{\mathrm{max}} = 8.9 \times 10^{-21} \left( \frac{10\mathrm{kpc}}{D} \right). \tag{13}$$

The GW signals from the collapsar phase are shown in Figure 7. At a distance of 10 kpc, those signals may be detected by aLIGO, CE, and ET.

In the central engine phase (hyperaccretion phase), the disk can emit GWs by anisotropic neutrino emission [131,133] and by precession [130]. For GWs from anisotropic neutrino emission, the typical frequency is at 1–100 Hz. As displayed in Figure 6, the progenitor mass and metallicity have little influence on GW signals, and the maximum amplitude of GWs from NDAFs is roughly $h_{\mathrm{max}} = 5 \times 10^{-22}$. GWs from disk precession have been studied by some previous works [130,178]. As mentioned above, Sun et al. (2012) [130] studied the GWs from the BH-disk precession system. The disk-driven jet precession may be common in all kinds of BH accretion systems since the only necessary condition is that the angular momentum of the initial accretion flow is misaligned with the BH spinning axis [179]. In Romero et al. (2010) [178], they assumed the whole accretion disk precesses as a rigid body and calculated the GW emission. The GW signals from disk precession are expected to peak at tens of Hz and have comparable amplitudes to GW signals from the anisotropic neutrino emission. Besides, van Putten & Levinson (2003) [180] suggested another GW source of the central engine. They studied the GWs from a magnetized torus around a rapidly rotating BH, whose accretion is suspended because the BH–torus interaction can transfer angular momentum to the torus or the disk and prevent the accreted materials from falling into the horizon. The configuration of the accretion torus itself may develop to the large nonaxisymmetries, and a large fraction of energy of this system is released by GWs. The typical frequency of GWs from the suspended accretion is a few hundred Hz, and the amplitude from it is much larger than that from NDAFs.

In the GRB jet phase, the relativistic jets are expected to be GW sources [157,158,181,182]. Sago et al. (2004) [157] investigated GW emitted in the acceleration phase of the GRB jet based on the internal shock model. The ultrarelativistic nonspherically symmetrical acceleration of energetic jets would produce GWs. At a distance of 10 kpc, the max amplitude of GWs in the acceleration phase is $\sim 10^{-22}$ at the frequency of $\sim 0.1$ Hz. Akiba et al. (2013) [158] studied the GWs in the decelerating phase of GRB jets. In the decelerating phases, the kinetic energy of the jet is converted into the energy of gamma-ray photons and a burst of GWs would be produced if the emission is partially anisotropic. The typical frequency of associated GWs is at 10–100 Hz, and the amplitude is approximately $\sim 10^{-24}$ at 10 kpc, which is difficult to detect now. Thus, the GW signals from the GRB jet phase are likely to be detected at 0.1–10 Hz by DECIGO/BBO and ultimate-DECIGO.

The GW signals related to various mechanisms from three phases have different characteristic frequencies and amplitudes. Moreover, the collapsars phase occurs earlier than the central engine phase and GRB jet phase. One may detect high-frequency GW signals form the collapsar phase first and then the low-frequency GW signals from later phases. For a Galactic source, those GW signals are expected to be detected, which can help constrain the characteristics of collapsars and central engines.

## 5. Summary

In the era of multimessenger astronomy, collapsars are promising multimessegner transient sources. In this review, we mainly focus on the two essential physical processes in the collapsar model, i.e., jet propagation and BH hyperaccretion. We have reviewed

the theoretical progress in the jet propagation in collapsars and multimessenger signals of collapsars. Various progenitor stars, including WR stars and Population III stars, can produce different kinds of GRBs in the collapsar scenario. The jets can be powered by neutrino annihilation or the BZ mechanism. The outflows in collapsar would affect the BH hyperaccretion in collapsars and jet-driven SNe. The properties of GRBs from collapsars are determined by the jet propagation and BH hyperaccretion together. There is competition between the mass supply onto the BH hyperaccretion and jet propagation into the envelope, which are definitely dependent on the density profiles of the collapsars. As a result, collapsars with different progenitors can produce LGRBs, SGRBs, or failed GRBs. Meanwhile, the density profiles of collapsar also affect the isotropic energy of GRBs. Therefore, the duration and isotropic energy of GRBs can help constrain the progenitors of collapsars.

The typical neutrino luminosity of NDAFs in the center of collapsar is $10^{50}$–$10^{52}$ erg s$^{-1}$, peaking at 10–20 MeV. Those neutrino signals are expected to be detected by Hyper-K within 1 Mpc. The mass of a collapsar has little influence on the neutrino spectrum, and a low metallicity is beneficial to the production of low-energy ($\lesssim 1$ MeV) neutrinos. The neutrino emission of NDAFs is different from that of PNSs. For a Galactic collapsar event, the neutrino spectrum may help distinguish the remnants of the collapsar. Moreover, the neutrino physics, such as mass and its hierarchy, mixing, and oscillation, could be intensively investigated.

The typical frequency of GWs generated by the anisotropic neutrino emission from NDAFs in the collapsar scenarios is ∼1–100 Hz. Higher progenitor mass and lower metallicity are favorable for the GW radiation of NDAFs. The GWs from NDAFs might be detected by DECIGO/BBO, ultimate-DECIGO, ET, and aLIGO in the detectable frequency range of ∼10–100 Hz. Moreover, there are many GW sources in a collapsar event. We briefly summarized the GW emission in the different phases of collapsars (the collapsar, central engine, and GRB jet phases). Considering that the three phases occur in a time sequence, one may distinguish the detectable GWs from the different phases, which can partly verify the collapsar model and BH hyperaccretion solution.

In the future, GRBs, neutrinos, and GWs from collapsars might be detected together. Since all those signals are determined by the properties of progenitor stars, we may obtain the accurate and authentic properties of the progenitor stars and central BH accretion systems by combining the information of those signals.

**Funding:** This work was supported by the National Natural Science Foundation of China under grants 12173031 and 12221003.

**Data Availability Statement:** Not applicable.

**Conflicts of Interest:** The author declares no conflict of interest.

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
