# Peer review of "Black Hole Hyperaccretion in Collapsars: A Review"

_universe, doi:10.3390/universe8100529_

Round 1

Reviewer 1 Report

As a review paper, I think the manuscript is reasonably well written. I only have some comments on the matter of presentation. 

(1) SN is supposed to the abbreviation of supernova, but what is SNe? This acronym appears throughout the manuscript without being defined. 

(2) line165: The authors should at least briefly explain what high-z universe is.  

(3) line 320-321: "...r<r_ign emerges in the disk" : Do the authors want to say that r_ign becomes nonzero so that there are regions with r<r_ign? Please rewrite this part, because the phrase does not seem to make sense in its current form.

(4) lines 349-354: This part is written as if the authors are writing a research article and reporting the result for the first time. Since they are just referring to their work while writing a review, I think it is better that they refrain from using the term "we", and instead use more objective expressions such as "in  ref.[...], the authors did this and that" or some similar expressions. The revisions should be made to other parts of the manuscript where the authors cite their results. Too much details should be omitted, and their work should be treated on equal footings with other works. 

(5)  line 26: "...previous works, there are..." => "...previous works, which are..."

(6) line 62: "... collapsars mainly..." => "... collapsars are mainly ..."

(7) line 90:    jet launched -> jet is launched

(8) line 106-107: "A more detailed description of jet-cocoon model can see in Bromberg et al."=> "For a more detailed description of jet-cocoon model, see Bromberg et al."

(9) 113: "..modeling ... show..." => either "modelings ... show" or "modeling ... shows"

(10) 209: "Then the accretion rate decreases" => "When the accretion rate decreases"

(11) 278:  "The GRB tGRB" => The GRB duration tGRB

(12) 402: "cool to a cold and deleptonized NS" => "cool to form a cold and deleptonized NS"

(13) 707: "combing"=>"combining"

Author Response

Referee Report

We appreciate your helpful suggestions. The main revisions in the text are marked in boldface. Hopefully our paper can be accepted for publication.

(1) SN is supposed to the abbreviation of supernova, but what is SNe? This acronym appears throughout the manuscript without being defined.

Re: Thanks. SNe is the abbreviation of supernovae. We define it in line 41 now.

(2) line165: The authors should at least briefly explain what high-z universe is. 

Re: Thanks. We added some sentence in line 165 to introduce GRBs from high-z universe.

(3) line 320-321: "...r<r_ign emerges in the disk" : Do the authors want to say that r_ign becomes nonzero so that there are regions with r<r_ign? Please rewrite this part, because the phrase does not seem to make sense in its current form.

Re: Sorry for inappropriate statement. We rewrote some sentences in line 325-330 to introduce r_ign.

(4) lines 349-354: This part is written as if the authors are writing a research article and reporting the result for the first time. Since they are just referring to their work while writing a review, I think it is better that they refrain from using the term "we", and instead use more objective expressions such as "in  ref.[...], the authors did this and that" or some similar expressions. The revisions should be made to other parts of the manuscript where the authors cite their results. Too much details should be omitted, and their work should be treated on equal footings with other works.

Re: Thanks for reminding. We rewrote some sentences and deleted many details in the manuscript.

(5)  line 26: "...previous works, there are..." => "...previous works, which are..."

Re: Corrected.

(6) line 62: "... collapsars mainly..." => "... collapsars are mainly ..."

Re: Corrected.

(7) line 90: jet launched -> jet is launched

Re: Corrected.

(8) line 106-107: "A more detailed description of jet-cocoon model can see in Bromberg et al."=> "For a more detailed description of jet-cocoon model, see Bromberg et al."

Re: Thanks. We rewrote this sentence.

(9) 113: "..modeling ... show..." => either "modelings ... show" or "modeling ... shows"

Re: Corrected.

(10) 209: "Then the accretion rate decreases" => "When the accretion rate decreases"

Re: Corrected.

(11) 278:  "The GRB tGRB" => The GRB duration tGRB

Re: Corrected.

(12) 402: "cool to a cold and deleptonized NS" => "cool to form a cold and deleptonized NS"

Re: Corrected.

(13) 707: "combing"=>"combining"

Re: Corrected.

Reviewer 2 Report

See the attached pdf file.

About Manuscript ID: Universe-1954756

Dear Editor,

I have read this review article entitled with “Black hole hyperaccretion in collapsars: a review” in detail.

As is well-known, the core collapse of massive stars, and the coalescence of compact object binaries are believed to be powerful sources of gravitational waves (GWs) as progenitors of gamma-ray bursts (GRBs). In the collapsar scenario, a rotating stellar-mass black hole (BH) surrounded by a hyperaccretion disk might be active in the center of a massive collapsar, which is one of the plausible central engines of long GRBs. Such a BH hyperaccretion disk would be in a state of a neutrino-dominated accretion flow (NDAF) at the initial stage of the accretion process; meanwhile, the jets attempt to break out from the envelope and circumstellar medium to power GRBs. In addition to collapsars, BH hyperaccretion systems are important sources of neutrinos and GWs.

A GRB initiates when the jets can be separated from the envelope and circumstellar medium. The theoretical development of the multi-messenger astronomy of the BH hyperaccretion in collapsar centers is summarized in this article. The primary themes are MeV neutrinos from NDAFs and jet propagation in collapsars. In this review, the major subjects considered by the Authors cover GWs from collapsars, jet propagation in collapsars, and MeV neutrinos from NDAFs and proto-neutron stars. Future detections could include GRBs, neutrinos, and GWs from collapsars. Since the characteristics of progenitor stars impact the characteristics of all those signals, combining the information from those signals can allow the scientists to identify the precise and real characteristics of progenitor stars and core BH accretion systems.

My views on this study are as follows: It is obvious that the Authors are experts in the subject. A good literature knowledge and bibliography from the past to the near future are included. The findings are depicted with the effective graphics. The possible future works are also included in the paper. And it's a typo-free i.e., a well written article. Therefore, my opinion is that the article should be accepted in its current form.

Author Response

Thanks for your positive comments.